# Gastrointestinal Parasites in Shelter Dogs: Occurrence, Pathology, Treatment and Risk to Shelter Workers

**DOI:** 10.3390/ani8070108

**Published:** 2018-07-02

**Authors:** Ali Raza, Jacquie Rand, Abdul Ghaffar Qamar, Abdul Jabbar, Steven Kopp

**Affiliations:** 1School of Veterinary Science, The University of Queensland, Gatton, QLD 4343, Australia; jacquie@petwelfare.org.au (J.R.); s.kopp@uq.edu.au (S.K.); 2Australian Pet Welfare Foundation, Kenmore, QLD 4069, Australia; 3Department of Clinical Medicine and Surgery, Faculty of Veterinary Science, University of Agriculture, Faisalabad 38040, Pakistan; agqamar2424@gmail.com; 4School of Veterinary Science, The University of Melbourne, Werribee, VIC 3030, Australia; jabbara@unimelb.edu.au

**Keywords:** animal shelters, dogs, nematodes, tapeworms, protozoa, treatment protocols, parasiticide resistance

## Abstract

**Simple Summary:**

Despite evidence of a minor role of gastrointestinal parasites in causing disease in owned pet populations prophylactically treated with anthelmintics, gastrointestinal parasitism remains an important consideration in the care of animals in shelters, and in owned pet populations in developing countries, where regular prophylactic treatment is lacking. In addition, the zoonotic potential of many organisms is a universal public health concern. Animal shelters facilitate spread of gastrointestinal parasites to incoming animals and shelter staff if there is overcrowding and frequent exposure to a contaminated environment. The prevalence of gastrointestinal parasites in shelter dogs is typically higher than in owned dogs. In this review, we report the prevalence of parasites in shelter dogs worldwide, and review parasite control strategies for use in shelters. We also discuss whether the shelter environment might magnify risks for development of parasiticide resistance in resident parasite populations. We recommend an integrated parasite control approach based on sanitation measures to reduce environmental contamination and accompanied with appropriate use of anthelmintics in shelter dogs. Ideally, every animal should be treated after fecal examination for parasites during its stay in the shelter, although it is recognized that for many shelters, the resources to do this might be prohibitive.

**Abstract:**

Dogs entering shelters can carry gastrointestinal parasites that may pose serious risks to other animals, shelter staff and visitors. Shelters provide an environment that could facilitate the spread of parasitic infections between animals. Nematodes and protozoa that transmit through ingestion or skin penetration are major enteric parasites of concern in shelter settings. *Ancylostoma* spp., *Uncinaria stenocephala*, *Toxocara canis*, *Toxascaris leonina*, *Trichuris vulpis* and *Dipylidium caninum* are the major helminths while *Giardia*, *Cryptosporidium*, *Isospora* spp. and *Sarcocystis* spp. are the most prevalent protozoan parasites in shelter dogs. The prevalence of gastrointestinal parasites in shelter dogs is typically higher than in owned dogs. A range of cost-effective drugs is available for prevention and control of helminths in shelters, notably fenbendazole, pyrantel, oxantel, and praziquantel. Parasiticide options for protozoan parasites are often cost-prohibitive or limited by a lack of veterinary registration for use in dogs. Environmental control measures reliant upon hygiene and facility management are therefore a mainstay for control and prevention of protozoan parasites in shelters. This philosophy should also extend to helminth control, as integrated parasite control strategies can allow anthelmintics to be used more sparingly and judiciously. The purpose of this article is to comprehensively review the current knowledge on the prevalence of gastrointestinal parasites most commonly found in dogs in shelters, canvass recommended treatment programs in shelter dogs, and to explore the likelihood that parasiticide resistance might emerge in a shelter environment.

## 1. Introduction

The major goal of animal shelters is to provide a temporary home for stray, lost or owner surrendered animals, until they can be reclaimed by the owner or rehomed (adopted) [1]. Shelter animals face various stressors, including overcrowding or isolation, unaccustomed environment and noise, limited physical activity and changed diet [2]. Due to these stress factors, and the combination of daily admissions of dogs from diverse origins, and the difficulty of preventing environmental contamination with infectious parasitic developmental stages, shelters provide favorable conditions for the establishment and spread of gastrointestinal (GI) parasitic infections. This is particularly for those infections transmitted by direct contact or ingestion of contaminated food, water or soil, and from licking contaminated surfaces [3]. In addition, length of stay in shelters ranges from a day to years [4], and therefore, some dogs can reinfect themselves, as well as infecting newly admitted dogs. As intakes are higher from lower socioeconomic areas [5], it is highly likely that incoming dogs may have had less than optimum parasite control [6]. Some of the enteric parasites of pet dogs and cats can affect humans through contamination of the environment by infectious larvae and eggs and through close contact with animals, however, a large number of owners are unaware of this unseen risk to public health [7,8]. 

Intestinal parasites can pose serious health problems in dogs, especially puppies [9,10] including retarded growth, lowered immune response to infectious diseases and generalized ill health [11]. For example, hematophagous parasites, such as hookworms, can cause traumatic lesions to the host’s intestinal mucosa resulting in anemia, which in puppies might prove fatal if not treated promptly [12,13]. While contemporary research has tended to focus upon extra-gastrointestinal parasites such as heartworm [13], gastrointestinal parasites remain a significant threat to animal health in shelter environments. There have been a few studies (limited to geographical locations) reporting the prevalence of gastrointestinal parasites in shelter dogs. This review article aims to comprehensively summarize the studies reporting the prevalence of parasites in shelter dogs worldwide, to explore parasite control strategies in a shelter environment, and to consider whether the unique environment of a shelter might magnify risks around the development of parasiticide resistance in resident parasite populations.

## 2. Magnitude of the Shelter Dog Population

Australia has one of the highest pet ownership rates of any country, with an average of 38% households owning a dog [14], which is higher than the United Kingdom (UK; 24%) [15], although slightly less than the dog ownership rate in the United States of America (USA; 48%) [16]. In Australia, in 2012–2013, approximately 211,655 dogs were admitted, at a rate of 9.3 admissions/1000 residents, to animal shelters and local government animal facilities (pounds). Of these admissions, the number of dogs reclaimed, rehomed or euthanized was estimated at 4.4, 2.9 and 1.9/1000 residents, respectively [17]. 

In the USA, there are more than 3500 animal shelters with an estimated 4.1 million dogs admitted in animal shelters in the year 2016 [18]. Although 2.3 million dogs were rehomed from shelters, only 30% of dogs were returned to their owners [19]. Animal shelters in the UK admit lower but still considerable numbers of stray dogs. A survey showed that in a 12 month period in 2010–2011, 126,176 dogs were admitted by the local councils across the UK [20]. Therefore, in western countries, a proportion of the dog population transits through or resides in, animal shelters and municipal pounds every year.

## 3. Parasites in Shelter Dogs

The shelter environment is conducive to parasite transmission in dogs, and those transmitted via ingestion of parasitic developmental stages from a contaminated environment are the major pathogens responsible for illness in shelter dogs [21]. Studies comparing different dog populations (stray dogs, owned dogs, kenneled dogs and shelter dogs) found that shelter and kenneled dogs carry gastrointestinal parasites more frequently compared to owned dogs. The higher prevalence in shelter dogs was attributed to increased exposure to parasites as a result of daily admissions of dogs from diverse origins, environmental contamination, and exacerbated by potential immunocompromise of the dogs due to various stressors in the shelter environment [7,22,23,24]. A number of studies reporting the prevalence of GI parasites of shelter and stray dog populations found the highest prevalence was 98% in Mexico [25] followed by 75% in Serbia [26], and 66% in Iran [24], with lower prevalences in Ethiopia (51%) [27], Malaysia (48%) [28], Portugal (39%) [8], Venezuela (36%) [29], Australia (37%; 35%) [7,22], and Canada (21%) [23]. 

The most frequently reported parasites in shelter dogs are helminths and protozoans [7,12,30]. Hookworms (*Ancylostoma* spp., *Uncinaria stenocephala)*, ascarids (*Toxocara canis*, *Toxascaris leonina),* whipworm (*Trichuris vulpis)* and the flea tapeworm (*Dipylidium caninum)* are the major helminths, while *Giardia*, *Cryptosporidium*, *Isospora* spp. and *Sarcocystis* spp. are the most prevalent protozoan parasites in shelter dogs. Table 1 outlines the prevalence of various parasite species in shelter dogs in various locations worldwide. A common limitation of these studies is reliance on microscopic examination of feces to diagnose different parasites. As far as nematodes are concerned, microscopy is generally a reliable method for routine diagnosis, however, the technique lacks sensitivity when compared to contemporary modalities, particularly polymerase chain reaction (PCR). Visual microscopy poses even more significant limitations for detection of protozoan parasites, as they are typically represented by very small sized cysts or oocysts in feces, which are difficult to reliably detect visually.

### 3.1. Nematodes

Depending on the mode of transmission, nematode infections in dogs can be divided into two types: (a) common nematodes that are easily transmissible in kennel-like arrangements and (b) unusual types that are transmitted directly and are mostly infectious at the time of excretion from infected animals [6].

#### 3.1.1. Hookworms

Hookworm species including *Ancylostoma caninum*, *A. braziliense*, *A. ceylanicum* and *Uncinaria stenocephala* are among the major intestinal pathogens of dogs [32,37]. *Ancylostoma* spp. primarily occur in moist and warm climatic zones; whereas *U. stenocephala* prefers temperate and subarctic regions [6,38]. Hookworm infection can occur either through ingestion or skin penetration of infective larvae. The larvae that penetrate the skin migrate through tissues and reach the lungs via the lymphatic system. In the lungs, these larvae infiltrate the alveoli and migrate up to trachea, from where they reach the intestine [39]. Trans-mammary transmission of *A. caninum* is considered to be the most important route by which puppies become infected, and the importance of the trans-placental transmission is still unclear [39,40].

Hookworm infections are more common in those groups of dogs (such as in kennels, pet shops and shelters) where the environment is contaminated due to a large number of dogs and accumulation of feces. The problem is more severe in unpaved areas where soil provides protection to hookworm larvae and makes sanitation processes ineffective [6]. The problem is greater in the shelter and kenneled dogs due to more frequent exposure of animals to feces, and is exacerbated during summer and wet seasons when temperatures and humidity are increased. The rapid hatching and development of hookworm larvae during periods of high humidity and temperature means that prompt removal of feces from the environment is essential to minimize transmission.

Consequences of hookworm infection depend on the host susceptibility, including age and health status, and on the worm burden and virulence of the species responsible for infection. The number of hookworms in a dog reflects the degree of exposure to a contaminated environment, and level of contamination, which is associated with other factors, including temperature, the substrate (sand or soil) and moisture that aid the development of parasite larval stages [6,39]. The signs of hookworm infection depend on the pathogenicity of the species, with clinical signs ranging from none to rapidly developing depression and death. For example, *A. caninum* is more pathogenic in dogs than *A. braziliense* and *U. stenocephala* [41], relating largely to the amount of blood ingested by each feeding worm. Young puppies acquire infection directly from their mother via milk and per-acute signs may occur including lethargy or depression, pale mucous membranes, and soft liquid bloody feces. Worms do not shed eggs until the end of the second week of infection, making early diagnosis difficult. Older pups may develop acute infection when exposed to a large number of infective larvae. Fluid loss and malabsorption result in diarrhea (often frank melena), pale mucous membranes, rough hair coat and failure to thrive [13,42]. Infected animals show a significant reduction in hemoglobin levels, packed cell volume and total erythrocyte counts [11]. Chronic hookworm infections in older dogs are usually subclinical and can be diagnosed by fecal examination only, although immunosuppression may result in clinical illness. Iron deficiency anemia may predominate in such animals. Infections that occur in animals in poor health due to other conditions, such as malnourishment and emaciation, are termed secondary hookworm disease and result in prominent anemia in addition to signs such as emaciation, or other specific signs associated with the underlying cause of ill health [6]. 

Canine hookworm infections can pose serious health hazards to humans, including cutaneous larval migrans and eosinophilic enteritis [40]. Although all canine hookworm species have the potential to migrate cutaneously in humans, *A. braziliense* is the most commonly involved hookworm in this condition [43]. When other species are responsible, milder creeping eruption or ‘ground itch’ is more likely, representing a localized dermatitis. Creeping eruption or cutaneous larval migrans is more prevalent in people who are regularly in contact with contaminated ground or surfaces contaminated with feces, particularly when they have exposed skin [44]. Shelter staff, particularly people involved with cleaning, are at risk of acquiring hookworm infection from the shelter environment. Lesions start at the penetration site and extend as an erythematous papular or vesicular rash [37]. Parasites can move up to several inches a day, and their movement is more rapid at night [6]. Raised lines indicate the course of larval movement which sometimes results in bullae formation. Crust formation results from drying of the surface of the lesion which may produce skin irritation. Sometimes itching due to infection becomes more intense or intolerable in sensitive patients [45].

*Ancylostoma caninum* is generally accepted to be responsible for human eosinophilic enteritis and has been associated with cases reported from Australia and the USA, based on clinical signs and seropositivity of patients. *A. caninum* is adapted to ingestion or skin penetration as a mode of transmission, however, based on experimental infection of humans with *A. caninum* larvae, ingestion of infective larvae is the major route for the acquisition of eosinophilic enteritis. Larvae can be ingested through contaminated water or food [37]. Signs include abdominal pain, eosinophilia and eosinophilic infiltration of the bowel wall [46]. Therefore, it is important that shelter workers adopt proper hygiene, including hand washing and wearing personal protective equipment, to avoid contact with hookworm larvae.

#### 3.1.2. *Toxocara canis*

*Toxocara canis*, a roundworm infecting dogs has worldwide prevalence (Table 1), although this has decreased significantly over time, presumably due to routine use of broad-spectrum anthelmintics [7,40]. It is more prevalent in puppies and can be fatal, especially when there is heavy prenatal infection [47]. Eggs of *T. canis* are very resistant and can withstand harsh environmental conditions [48]. Infection occurs via ingestion of eggs containing infective 3rd stage larvae; transplacental infection is another major route of *T. canis* infection, and puppies can also be infected through the transmammary route [40,48]. After ingestion, larvae hatch from the egg, penetrate the intestinal mucosa and enter into the portal circulation. After reaching the lungs via the pulmonary artery, the larvae either undergo tracheal migration and develop to sexual maturity in the intestine, or undergo somatic migration to remain arrested in the extra-gastrointestinal tissues of the dog [40]. The probability of tracheal migration is higher in puppies and decreases with age, while that of somatic migration is higher in adults, particularly female dogs. Arrested larvae in females can be activated during the last trimester of pregnancy and transferred to pups *in utero* before whelping and through milk during lactation [6,40].

Heavily infected pups may continuously whine, shriek and adopt a particular posture by overlapping hind limbs while standing or walking. Worms respond to irritant stimuli such as an acidic pH and entangle into knots, which can result in intestinal obstruction and/or rupture, bile duct obstruction and epileptiform seizures, and ultimately death [6,40]. Other clinical signs vary with age and health status of the dog, and stage and severity of infection-larval migration through the lungs can result in a cough, nasal discharge, pneumonia and edema of the lungs, while adult worm activity may produce mucoid enteritis, vomiting, diarrhea, ascites, anorexia, anemia, emaciation, poor body coat and pot belly, if the worm burden is heavy [13,39,40].

Human toxocariasis is present almost all over the world and is the most common zoonotic parasitic infection from pets [49]. Human toxocariasis manifests as a number of syndromes including visceral, ocular, neural and subclinical forms, with most infections subclinical [50]. Humans can acquire infections in a number of ways including ingestion of contaminated soil (frequent in toddlers), ingestion of partial or whole paratenic hosts such as raw liver of domestic animals, or ingestion of uncooked vegetables, particularly those fertilized or contaminated by excreta of infected animals. A recently suggested route of infection is contact with embryonated eggs on the dog’s hair coat [51]. *T. canis* eggs are not infective when passed into feces and require three to six weeks to become embryonated, depending on the environmental conditions. Therefore, eggs attached to the hair coat are less likely to be infective [47]. 

Visceral larva migrans is caused by migration of larvae through host tissues and is associated with fever, abdominal distress and pain, eosinophilia, leukocytosis, hepatomegaly and some respiratory signs (bronchitis, asthma or pneumonia) [48]. Visceral larva migrans is more severe and frequently reported in children from one to three years of age, which is likely due to the frequent exposure of children to the soil and potential for ingestion of contaminated soil [47].

Ocular larva migrans is caused by migration of larvae to the eye that results in damage to the eye and optic nerve. Ocular larva migrans usually occurs without signs of visceral larva migrans in children aged five to ten years [49,52]. It is manifested by impaired eyesight to total blindness due to endophthalmitis, retinal granulomas, and detachment of the macula [53]. For shelters which do not follow strict disinfection and cleaning schedule, shelter staff and visitors should be advised to wear personal protective equipment including disposable clothing and closed shoes to avoid contact with infective stages. The authors could find no reports of studies involving shelter staff to determine the prevalence of human hookworm or *Toxocara* infections, and it is suggested that this should be a future avenue of investigation.

#### 3.1.3. *Trichuris vulpis*

*Trichuris vulpis*, also called whipworm due to its body shape, is a nematode present in the large intestine of dogs, although the location varies from the cecum to colonic mucosa, depending on the worm population [6,54]. Dog whipworm is a ubiquitous parasite and found all over the world in kenneled, household, stray and shelter dogs [54]. Eggs are difficult to eliminate and are killed when exposed to extreme conditions, for example, dehydration and sunlight for an extended period. Canine infections occur through ingestion of embryonated eggs from contaminated soil or water. Infective larvae emerge from embryonated eggs and pierce the intestinal glands where they molt before colonizing the large intestine [54,55]. Eggs serve as a constant source of infection once mixed with soil because they are difficult to eliminate, and continually expose dogs to re-infection. Thus, the incidence and parasite burden of trichuriasis is higher in adult dogs compared to younger animals. The second factor that supports the higher prevalence in adult dogs is the absence of transmammary or transplacental routes of transmission [56]. Parasitic infection would be expected to be higher in shelters where there is contaminated soil in dog runs.

Infection in dogs ranges from subclinical to a range of gastrointestinal signs, although it is generally considered less pathogenic than hookworms and *T. canis*. Some dogs appear to tolerate a relatively high parasitic burden without showing clinical signs. Another factor that favors reduced pathogenicity is the slower development of the parasite [54]. Clinical signs in puppies include weight loss, reduced growth rate and predisposition to secondary pathogens. The penetration of the cephalic end of the parasite in the mucosa of the colon or caecum results in acute or chronic inflammation, which is more severe in young animals with heavy worm burden [55]. Typical clinical symptoms are characterized by alternative episodes of mucoid, watery or hemorrhagic diarrhea with a period of normal feces, together with weight loss and anemia in more severe cases [55].

Although *T. vulpis* is closely related to human whipworm *Trichuris trichiura*, the zoonotic importance of *T. vulpis* as a cause of human disease is still controversial. There are few case reports describing *T. vulpis* as a cause of disease in humans [57], and the diagnosis was based on egg measurements of both these species without the use of any molecular diagnostic technique, therefore the zoonotic potential of *T. vulpis* requires further confirmation.

#### 3.1.4. *Strongyloides stercoralis*

*Strongyloides stercoralis,* a thin threadlike nematode of dogs, humans and other canids [58] that lives entangled in the mucosa of the small intestine. The parasite has significant zoonotic potential and dog handlers or caretakers can acquire infection from dogs under their care [59]. The parasite is transmitted by penetration of infective larvae through the skin in both human and dogs [60]. Larvae migrate to the lungs after skin penetration, are coughed up, swallowed and develop in the small intestine to adult female worms which produce eggs. Eggs hatch within the mucosa of the intestine and first stage larvae are passed into the feces where they develop to the infective larval stage, or can mature to infective larvae in the gut and autoinfect the host after lung migration [61]. Larvae are also passed through milk to puppies. The disease varies from subclinical to clinical signs including gastrointestinal signs such as diarrhea, and a variety of signs as a result of damage to lungs and other tissue by migrating larvae [62]. It appears that young dogs especially puppies are more prone to developing clinical strongyloidosis, and *S. stercoralis* infection was associated with the death of a 10-week old puppy in a kennel [63]. Damp areas are more favorable for the persistence of larvae, thus heavily soiled cage areas are highly conducive to harbor larvae of *S. stercoralis* [61]. A recent study has described the occurrence of *S. stercoralis* infection in shelter dogs in Southern Italy [64], representing a potential risk for staff, visitors, and sheltered animals. Importantly, a single course of treatment with fenbendazole alone, or combined with moxidectin and imidacloprid spot-on was not effective in eliminating infection in two out of six dogs, suggesting that control strategies targeting the environment should be implemented to reduce the risk of infection.

### 3.2. Protozoa

#### 3.2.1. *Giardia* spp.

*Giardia* is a pathogenic intestinal protozoan, which causes disease in animals and humans. *Giardia intestinalis* (syn. *G. duodenalis* or *G. lamblia*) is the single species responsible for disease in mammals, including humans. Genotyping of *G. intestinalis* has classified *Giardia* into different assemblages, which infect certain groups of hosts. Assemblages A/B infect humans and other animals including dogs, cats, and livestock, assemblages C/D infect dogs, while assemblages E, F and G infect hoofed stock, cats, and rats, respectively [65,66]. The assemblages A/B were also reported from dogs and cats, which means that humans can acquire infection (assemblages A/B) from dogs. *Giardiasis* in pet dogs is reported throughout the world with varying frequencies including 16% in USA [67], 22% in Australia [7], 23% in Portugal [8], 59% in Hungary [68] and 13% (detected by copro-antigen) and 64% (detected by PCR) in Canada [69,70]. The prevalence of *G. intestinalis* in the shelter and kenneled dogs is also worldwide (Table 1). A survey of different dog populations for the presence of *Giardia* reported a prevalence of about 10% in well cared for dogs, 36%–50% in puppies and 100% in dogs in breeding establishments and kennels [71]. The risk of *Giardia* infection is increased with increased frequency of anthelmintic administration. This is likely due to the vacation of a niche in the intestine caused by the action of anthelmintics on major parasites, including hookworms and ascarids [7].

The life cycle of *Giardia* is direct and the transmission occurs via ingestion of infectious cysts from surfaces or in the soil, food, or water that has been contaminated with feces from infected animals. *Giardia* has two developmental stages; cyst and trophozoite. Ingestion of cysts is followed by excystment in the duodenum as a result of exposure to gastric acid and pancreatic enzymes. Two trophozoites are released after excystment of each cyst, which attach at the base of proximal small intestine villi and absorb nutrients from the intestinal lumen through their cell membrane. Trophozoites reproduce by binary fission to increase the number of parasites. Some protozoa undergo encystment and pass out in the feces because unprotected trophozoites cannot transmit infection and die in the environment [72].

The spectrum of disease varies from subclinical to overt clinical signs depending upon age and nutritional status of the animals and other comorbidities. Most infected immune-competent dogs act as carriers without showing overt clinical signs [73]. When dogs develop clinical disease, the most consistent clinical feature of giardiasis is diarrhea, which may be acute or chronic, self-limiting and intermittent, or continuous and lead to dehydration. Infected animals may develop severe enteritis leading to maldigestion and malabsorption as a result of the host’s inflammatory response. Some dogs may develop malodorous diarrhea, steatorrhea (due to indigestion of fat), weight loss and stunted growth [67,72].

Although most human cases of *Giardia* are not directly acquired from animals, dogs carry strains of *Giardia* that are potentially pathogenic to humans and are transmitted through hand-to-mouth transfer of cysts from infected feces or fecally contaminated surfaces or ingestion of contaminated food or water. The zoonotic potential of dogs infected with *Giardia* depends on genetic diversity that occurs within *G. intestinalis* species. Most of the linkages of *G. intestinalis* seem to be host specific or have limited host range [65]. Dogs are carriers of A-I and B assemblages transmissible to humans. Historically, A-II were categorized as human isolates, but recently genotypes of A-II assemblages have been isolated from dogs in India [41] which reflect their potential zoonotic potential. Based on the analysis of genetic data, the major assemblages of *Giardia* are quite different to each other, suggesting that assemblages of *G. intestinalis* should be divided into different species by revising the taxonomy of *Giardia* [65,74].

The symptoms of giardiasis in humans include chronic diarrhea, dehydration, weight loss, abdominal pain, nausea, and vomiting. The frequency and severity of symptoms can be highly variable with many individuals remaining free of clinical signs. Giardiasis in humans can be prevented by adopting proper sanitation and hygienic measures to minimize the risk of transmission [24]. *Giardia* can be diagnosed by fecal flotation method [69] but molecular techniques including enzyme-linked immunosorbent assay ELISA [31], PCR [68,69] and SNAP *Giardia* test [67] are more sensitive and are recommended. Diagnosis of *Giardia* by identification of cysts in feces will lead to false negative results due to their small size and intermittent shedding in the feces. 

#### 3.2.2. *Cryptosporidium* spp.

Dogs most commonly harbor *Cryptosporidium canis* while other species of the genus *Cryptosporidium* are found in the gastrointestinal tract of cats, humans and other animals [75]. However, *C. parvum* infects humans and other mammals, including dogs, hence its zoonotic importance [22]. Reports of the prevalence of *Cryptosporidium* in dogs are limited and range from 0% to 53% throughout the world [34,76]. Dogs acquire infection through the direct route via ingestion of food, water or soil contaminated with feces containing oocysts, or licking contaminated surfaces. *Cryptosporidium* cysts are immediately infective after passing in feces [77], facilitating shelter staff and owners acquiring infection from infected dogs. Since these parasites transmit through ingestion of oocysts from an environment contaminated by infected feces, it would be expected to be more prevalent in crowded conditions like shelters, kennels and pet shops, especially if conditions are unsanitary [78]. *Cryptosporidium* has been reported from shelter dog populations in various countries (see Table 1).

Healthy dogs usually remain asymptomatic and *Cryptosporidium* infection is mostly self-limiting. Acute onset of infection occurs due to the short prepatent period (4–5 days) of the parasite. Clinical disease is manifested by small bowel diarrhea and straw-colored feces and occurs in immunocompromised animals. Thus younger puppies, older dogs or dogs with concomitant infections are at higher risk [76].

There have been studies reporting the association of *C. canis* with human infections in developed countries [75]. Immunocompromised people and malnourished children are at risk of clinical disease [76]. In humans, cryptosporidiosis commonly manifests as watery diarrhea, abdominal cramps and pain, although symptoms mostly resolve spontaneously. In immunocompromised individuals, the infection can become chronic, and leads to malabsorption and occasionally death [79]. Diagnosis of *Cryptosporidium* is difficult through the conventional method of fecal examination because of the small oocysts, and microscopy is not a reliable method to diagnose *Cryptosporidium* cysts in feces. Molecular techniques, including fluorescent antibody assay [6], ELISA [34] and PCR [79], should be used to diagnose cryptosporidiosis.

#### 3.2.3. *Isospora* spp.

Three major species of *Isospora* infecting dogs are *I. canis*, *I. ohioensis* and *I. burrowsi* while *I. neorivolta* is less commonly reported. *Isospora* has worldwide distribution and appears to be host specific [80]. The three species other than *I. canis* are not distinguishable on the basis of oocysts in feces and are termed *I. ohioensis-complex* [81]. A study in the USA reported a prevalence of 5% in shelter dogs [82] while *I. canis* has also been reported in the Australian owned dog population (7% and 1.4%) [7,32]. In the life cycle of *Isospora,* unsporulated oocysts are passed in the feces followed by sporulation in 9–12 hours in a favorable environment. Infection occurs following ingestion of sporulated oocysts or through an indirect route via ingestion of an infected paratenic host such as rodents. Insects may also serve as vectors to transmit sporulated oocysts [80].

Infection with *Isospora* usually remains subclinical, but clinical signs may develop in young animals. Infection with *I. canis* and *I. ohioensis* can be associated with mild to severe large intestine diarrhea, abdominal pain, vomiting and general malaise. Severe infection can lead to dehydration and death in younger animals. Moderate intestinal damage may lead to retarded growth in puppies, even when gastrointestinal signs are absent [83]. *Isospora*, unlike *Cryptosporidium*, is not as difficult to diagnose using microscopy because oocysts are large and often numerous in feces [81]. Therefore, *Isospora* infected animals usually can easily be identified using microscopy in shelter settings.

#### 3.2.4. *Sarcocystis* spp.

*Sarcocystis* has an obligatory heteroxenous (two hosts) life cycle involving herbivores as intermediate hosts and carnivores as definitive hosts. There are at least 21 species of *Sarcocystis* that are found in dog feces [84]. The major species infecting dogs are specific depending on the intermediate host involved including the cattle-dog (*S. cruzi*), the sheep-dog (*S. tenella*), the goat-dog (*S. capracanis*), the horse-dog (*S. bertrami*) as well as some other species [85,86].

Fully sporulated oocysts and sporocysts are excreted in host feces with no development in the external environment. Normally dogs acquire the infection by eating the flesh of infected herbivores. The sporocysts undergo gametogony (the development into male and female gametes), fertilization and sporulation in the dog followed by shedding of sporulated oocysts in feces of infected dogs. These infect herbivorous hosts after ingestion of vegetation contaminated by feces with sporulated oocysts. Stages of asexual reproduction, called schizogony and encystment occur in herbivores [6,86]. 

Dogs usually show no illness, although fever, lymphopenia, thrombocytopenia and myositis accompanied with reluctance to move, generalized pain and muscle wasting has been reported [87]. *Sarcocystis canis* is also responsible for severe hepatitis, encephalitis, dermatitis, and pneumonia in dogs, particularly in puppies [85]. However, *Sarcocystis* spp. that affect dogs are not zoonotic. 

### 3.3. Cestodes

*Dipylidium caninum* is the most common intestinal tapeworm of dogs, which is transmitted by fleas and biting lice [88]. The mature segments of *D. caninum* have the appearance of cucumber seeds. Eggs accumulate in packets formed by out-pocketing of the uterine wall in the segment [6]. Cysticercoids (a larval form of tapeworm) of *D. caninum* develop in fleas or lice and are ingested by dogs, especially fleas during grooming. Within dogs, cysticercoids take two to three weeks to develop into segment-shedding tapeworms in the small intestine, and anthelmintic therapy should be accompanied by flea control programs [89]. *Dipylidium caninum* is present in dogs throughout the world [30,90]. Tapeworm infection is usually asymptomatic in dogs and cats, but a heavy infestation can lead to poor growth and intestinal obstruction in puppies. The migration of tapeworm segments may cause anal pruritis manifested by scratching of the perianal region against the wall or scooting on the ground [6]. In animal shelters, diligent flea control will minimize transmission of *D. caninum*. The infection has also been reported in humans, most commonly children [91,92] associated with diarrhea and abdominal pain [91]. 

## 4. Treatment Programs for Gastrointestinal Parasites in Shelter Dogs

### 4.1. Recommended Standards of Treatment

Shelter medicine requires a ‘herd health’ approach to control, manage and reduce the transmission of disease. Because of the numbers of animals involved, cost-benefit considerations are important influencers of management choices [6]. Although gastrointestinal parasites are common in dogs entering shelters, the cost of regularly administering parasiticides to all dogs is often prohibitive. A recent survey reported that most dogs were treated only once, usually at the time of entry, and often only adoptable dogs were treated [93]. Dogs that harbor parasites do not always show clinical signs, and left untreated, contaminate the environment facilitating transmission to other dogs, and increase the risk of infection to shelter workers, visitors and new owners [94]. In a study of nine shelters in Louisiana USA, the “anthelminthic protocols for the studied shelters in Louisiana State were found to be inadequate and needed to be revised to prevent the spread of gastrointestinal parasites” [93]. Animal shelter management has a responsibility to minimize the risk of parasite spread to other dogs and humans.

According to the Guidelines for Standards of Care in Animal Shelters [95]:“Animal **should** receive treatment for internal and external parasites prevalent in the region and for apparent parasitic infection harbored by the animal at the time of entry”.“Ideally, every animal **should** be dewormed on entry and regularly throughout their stay in a shelter, but at a minimum, animals **must** receive anthelmintic drugs against roundworms and hookworms before leaving the shelter.”

Optimum anthelmintic control assists shelters in maximizing the chances of adoption because dogs that are obviously in poor condition or with diarrhea are less likely to be adopted, and the cost of treatment to owners may pose a barrier to adoption. It also reduces the risk to other dogs in the shelter, as well as to shelter staff and visitors from zoonotic parasites. Considerations for treatment include frequency, target animals, and anthelmintic choice, in the context of the type of GI parasites being encountered at the shelter, cost of drugs and labor, and if there is evidence of resistance to anthelmintic drugs. As soon as possible after entry to the shelter, to minimize environmental contamination and risk of infection to shelter workers, all dogs should receive anthelmintic treatment. However, if there is evidence of resistance to anthelmintic drugs, other strategies are will need to be considered. These include treatment tailored to the outcome of fecal examinations, instead of treating all the dogs in a shelter on an empirical basis. Such a strategy can assist in managing the cost of parasite control in a shelter, when more expensive anthelmintics may need to be considered. Note it is not recommended that fecal examinations only be performed on dogs with diarrhea, because the substantial fecal shedding of eggs can occur without clinical signs. Signs of diarrhea resolve in many dogs within several days, although egg shedding continues and then becomes more sporadic once the infection is well established. Importantly from a diagnostic perspective, most of the overt clinical signs are seen during the establishment phase of infection, where there is the onset of clinical signs but no shedding of eggs in feces. 

Selective treatment regimens, such as the concept of targeted selective treatment, are favored in the livestock industries to retain a proportion of untreated parasites in the population (termed parasite refugia). Such strategies call for the treatment of only severely burdened or clinically affected individuals, while clinically unaffected individuals with lower parasite burdens are not treated. The key driver of such strategies is the need to control parasites from an animal welfare and production efficiency standpoint, but to also spare entire parasite populations from regular treatments, which is known to be an important driver for selection of resistant parasite populations.

Selective treatment regimens are not typically an option in dogs, because many parasites harbored by dogs have implications for public health. It is not, therefore, appropriate to leave animals with lower parasite burdens untreated. These limitations could be problematic if significant parasiticide resistance were to develop in canine gastrointestinal parasite populations. Fortunately, parasiticide resistance has not yet been observed to be a significant issue in companion animal medicine. However, if resistance does arise, it is most likely to first emerge in shelter parasite populations, where parasite populations are exposed to regular parasiticide treatments and many dogs are potentially exposed to the same contaminated environment, especially where the length of stay is prolonged. For example, sanctuaries where dogs with behavioral or other issues may stay for years or their whole life. In shelters, particularly small and medium-sized shelters with a long length of stay, the number of dogs admitted on a daily basis may represent less inflow of parasites and hence genetic diversity, compared to large populations of parasites that dogs are exposed to in public areas such as parks. These issues are further developed and explored later in the paper. 

### 4.2. Non-Drug Control of Parasites

Non-drug control of parasites involves a number of strategies, including cleaning, good management, and appropriate husbandry practices. Alternative non-drug methods to control parasites are now imperative in many facets of large animal herd health because of the development of resistance against all classes of commonly used anthelmintics [96], although to date there are few reports of resistance in companion animals [97,98]. Cleaning and disinfection are the main strategies that can be used to reduce the prevalence and spread of parasitic infections in a shelter environment [99] and are considered the key to successful management. Ideally, feces should be cleaned immediately from outdoor areas, although the practicality of such a recommendation may vary with the availability of labor resourcing. [95,99]. Removal of feces assists in controlling parasites which are spread through environmental contamination, including nematodes, cestodes, and protozoa. Shelters that adopt good management practices, in addition to the strategic use of anthelmintics, are known to have a low prevalence rate of hookworms [93,100]. Proper cleaning involving removing feces and regular mechanical cleaning of cage floors, together with disinfection, decreases the load of parasitic developmental stages in shelters, which is a vital adjunct to parasiticide use [21,95,100]. From a design perspective, using concrete rather than dirt flooring for housing facilitates cleaning and reduces survival of parasites and hence environmental contamination. Cleaning and disinfection of any potential fomites including food and water bowls, shelter staff’s clothing, gloves, muzzles, and traps are also recommended to control the spread of parasites [95].

### 4.3. Drug Therapy and Drug Resistance

Shelter parasite control programs should be based on knowledge of the types of parasites in that specific geographical location, although, some common guidelines provided by the Association of Shelter Veterinarians are applicable to all shelter populations [95]. The general treatment protocol should include at least a dewormer effective against hookworms and ascarids. The nicotinic agonist tetrahydropyrimidines (pyrantel and oxantel) and benzimidazoles (notably fenbendazole and the pro-benzimidazole febantel) are the major classes of anthelmintics used in shelters. Pyrantel has a spectrum against only hookworm and ascarids; combination products containing both pyrantel and oxantel extend this spectrum to whipworm. For hookworm, both pyrantel and benzimidazoles have reasonably good activity against developing intestinal larval stages, although benzimidazoles have better activity against immature ascarids. Pyrantel alone has no effect on whipworm and needs to be combined with oxantel. Although benzimidazoles will treat whipworm, they have poor efficacy against immature stages. The extra burden of treatment for three consecutive days with benzimidazoles is an issue; offsetting this is the wider spectrum that extends to *Strongyloides* spp. (generally a less prominent parasite, however prevalence may be underestimated or unknown in many areas) and (at least to some extent) *Giardia*. However, regardless of the anthelmintic employed, three treatments at 2 to 3-week intervals is recommended to kill maturing immature stages, while the immature stages are more effectively killed with benzimidazoles. For animals housed long-term (i.e., for months), environmental control becomes crucial if a regular monthly worming regimen cannot be employed.

Puppies should be administered pyrantel pamoate every two weeks up to 16 weeks of age, and it is typically administered at the same time as vaccinations to decrease labor costs. In a well-managed facility with a minimally contaminated environment, this may be extended to every 3 weeks. Pregnant and nursing bitches should also be administered pyrantel pamoate every two weeks during their stay [13,39]. However, commercial compounds based on pyrantel but combined with other anthelmintics are often contraindicated in pregnancy, and compounds should only be used if specified to be safe in pregnant animals. Pyrantel offers protection against hookworm and ascarids, which are the two most important gastrointestinal helminths in puppies up to 16 weeks of age. The drug has a high therapeutic index and is very safe for use in young animals at the appropriate dose. Benzimidazole drugs also have a spectrum against hookworm, ascarids, and whipworm (Table 2). Activity against tapeworms of any kind requires incorporation of the isoquinoline praziquantel, or less commonly niclosamide. 

The most common drug for internal parasite control of nematodes in shelters is pyrantel pamoate administered orally [93,101]. Benzimidazoles are present in different formulations recommended for use in dogs and have good efficiency against a broad range of gastrointestinal parasites. Some broad-spectrum macrocyclic lactones (moxidectin, milbemycin oxime) are also effective in treatment and prevention of a wide range of parasites, including whipworms, ascarids, hookworms and some extra-intestinal nematodes as well [13]. However, macrocyclic lactone-based products are typically marketed for home environments; they are usually convenient topical ‘spot on’ formulations and can be cost-prohibitive for large-scale use in shelters or kennels. As evidenced by a recent study, 67% of treated dogs in shelters received pyrantel and 11% received a benzimidazole-based treatment [93]. Anthelmintics for use in dogs are shown in Table 2, which incorporates all commonly available classes, including those which may be cost-prohibitive in many shelter situations. Although metronidazole (25 mg/Kg every 12 h for five days) is the drug of choice for dogs with *Giardia* infection, it only eliminates infection in two-thirds of the cases. It has been suggested that for resolving clinical signs and cyst shedding metronidazole can be combined with fenbendazole (50 mg/Kg every 12 h for 3–5 days). However, this is a controversial approach. The authors are aware of anecdotal observations from some shelters that fenbendazole does not always appear to completely resolve *Giardia* infections. Further, there are concerns around the impact that frequent drug application might have with respect to magnifying resistance selection pressure. With respect to *Cryptosporidium*, infection is generally self-limiting in immunocompetent animals, lasting for 3 to 12 days and needs no specific treatment [102].

Generally, dogs respond quickly to anthelmintic treatment, and fecal egg counts decrease over time, for example, over 4–7 days for hookworm and ascarids [103]. Although quarantine of dogs for one week following treatment on admission would assist in decreasing the spread of parasites in the shelter, the increase in the length of stay would be of greater detriment to the dogs’ health and decreases the probability for a positive outcome. This is because increasing length of stay is associated with significantly increased risk of viral and bacterial diseases, and is not recommended for clinically healthy dogs [95].

In sanctuaries, where most dogs are expected to stay for a longer period (often for behavior modification), quarantine for 7 days is highly advocated to restrict the spread of gastrointestinal tract (GIT) parasites in the facility. Regardless, it must be recognized that resolution of shedding of parasitic forms following treatment is a progressive effect, and moving a treated animal to a clean environment soon after treatment regimen will not fully protect that environment from subsequent contamination. Due diligence with respect to prompt removal of feces, especially in the first seven to ten days following treatment, is an essential adjunct to treatment, together with use vigorous mechanical cleaning. Note that there is little evidence for the efficacy of accelerated hydrogen peroxide against hookworm larvae or ascarid eggs, although it is effective against parvovirus and other viral and bacterial agents, and hence is an integral part of the standard sanitation practice in many shelters. Infective larvae of hookworm are not killed by glutaraldehyde or 1% bleach but are very effectively killed by strong sunlight, 50ppm aqueous iodine, water above 80 °C (e.g., utilizing steam cleaning), 70% ethanol and Dettol^®^ (Reckitt Benckiser Group, Sydney, Australia). Strict use of sanitation (rapid removal of feces, mechanical cleaning and regular disinfection) also applies to quarantined animals, since reinfection from a contaminated environment will render the previous treatment ineffective with respect to breaking the parasite lifecycle. Ascarid eggs are much more difficult to kill because the developing larva is fully encapsulated (i.e., no free larval stage in the environment), hence the importance of promptly removing feces and thorough mechanical cleaning. 

It is recognized that routine use of anthelmintics, especially benzimidazoles, might hasten the development of anthelmintic resistance in canine helminths, as has occurred in livestock helminths where the benzimidazole class has been used heavily [104]. At this stage, the level of anthelmintic resistance in companion animals remains largely unknown, as definitive and widespread reports of resistance in canine helminths are lacking, and treatment efficacy is rarely monitored by veterinarians in private practice, or by shelter staff. In addition, the shelter environment may also contribute in slowing the development of anthelmintic resistance in companion animal parasites, as many animals entering a shelter likely have not been treated with anthelmintics or treated infrequently, and typically have a relatively short stay in the shelter (average approximately 30 days). This will dilute the resistant worm population, if present in frequently treated shelter animals, ultimately maintaining refugia of unexposed and drug-sensitive worms. Table 3 contains selected studies which highlight the ongoing efficacy of major drug classes employed to prevent and treat canine gastrointestinal parasites. Newer combinations of anthelmintics demonstrate higher efficacies than the anthelmintics commonly used in shelters, when evaluated in the field or controlled trials, generally in research or pet animals. However, studies should also be conducted in shelter environments to evaluate the efficacy of anthelmintics against resident parasite populations. While such populations may experience some degree of genetic influx with the turnover of dogs, shelters are a potential environment where a resistant fraction of parasites could develop over time (the same population of worms cycling through different dogs and being regularly exposed to anthelmintic treatment).

In the last four decades, the frequent use of anthelmintics has led to the development of resistance in livestock parasites to major anthelmintic classes [105]. Therefore, intensive use of anthelmintics in shelter animals, where husbandry conditions are more similar to that of livestock than pet dogs, could increase the population of resistant nematodes. Currently, there are limited reports about anthelmintic resistant nematodes of dogs. An isolate of *A. caninum* was reported from Brisbane, Australia, against which pyrantel was found to have an efficacy of just over 25% based upon a reduction in mean worm burden [97]. Further work by the same group observed phenotypic differences between high and low-level resistant isolates of *A. caninum*, where pyrantel efficacies in-vitro varied from 28% to 71% [101]. These observations formed the basis of a prospective laboratory test for detecting resistance to pyrantel in canine hookworm isolates. 

At this time, reports of anthelmintic resistance in dogs and cats appear to be confined to regions and limited in scope and repeatability across different parasite isolates. However, sporadic accounts do serve as a warning that it is indeed possible for resistance to arise in canine helminths, further highlighting the need for the responsible and judicious use of anthelmintics, and parasiticides generally, and the advisability of monitoring response to treatment.

Further research is required in other locations to clarify the extent of anthelmintic resistance in shelters. Findings of more widespread resistance will likely result in changes to recommendations of anthelmintic treatment protocols in shelters. For example, attention to ensuring correct dosing of dogs on a body weight basis and performing fecal examinations after treatment to monitor the efficacy of anthelmintics, and to identify to which drugs, and in which locations, there is the development of anthelmintic resistance. 

The slow emergence of drug resistance in companion animals is likely due to the difference in the management of pet animals, which unlike livestock, are kept individually or in small numbers, which minimizes spread of resistance when it occurs. In addition, the efficacy of treatment is not being monitored in the same way to detect resistance [101]. However, the dynamics in shelters and breeding kennels, where larger numbers of dogs are housed on the same site and managed and treated as a herd, are comparable to that of livestock.

Public health concerns about parasitic infections also need to be addressed. For instance, if resistance developed in parasites within shelter dogs and these dogs are subsequently adopted, these animals could harbor their own sub-population of resistant worms. These animals would soon establish their own isolated parasite community, which would be resistant to the available treatments [111]. It is well established that maintenance of refugia (unexposed and susceptible worms) is very important to slow the process of resistance development in livestock [112]. However, given the zoonotic risk to shelter staff, pet owners and potential of infection to other animals in the community, recommendations to deworm all dogs regularly might act as a driving factor for the development of anthelmintic resistance, due to a decreased proportion of parasites in refugia [113]. Therefore, shelter veterinarians should consider the potential for development of drug resistance while designing the parasite control programs for the shelter dogs. Regular fecal examinations following treatment should be used to monitor the efficacy of drugs. If resistance becomes evident, treatment programs based on a wider range of anthelmintics from different groups will need to be implemented, although the cost will be higher. Therefore, future research focusing on extending the range of available anthelmintics for use in companion animals is warranted, however, given the time and cost of developing new drugs, there is a need to manage the use of the existing drugs to preserve their usefulness for as long as possible. Box 1 has recommendations that animal shelters should consider adopting for specific drug and non-drug regimens for effective control of gastrointestinal parasitism in shelter dogs.

Box 1Recommendations for sanitation and anthelmintic regimes for use in shelters.➢On intake to the facility, administer an anthelmintic containing either pyrantel pamoate (ideally repeated every 2–3 weeks for 3 doses to kill maturing immature stages), or fenbendazole (50 mg/Kg body weight, every 24 h for three days). Fenbendazole clears *Toxocara* (80%–100%) and hookworm (99%–100%) infections within 9 to 16 days after treatment [105].
➢Fenbendazole is also reported to have good efficacy against *Giardia* spp. [114], and *Giardia* cysts disappeared after three days of treatment for three consecutive days [115]. If *Giardia* spp. are an issue in incoming dogs, use fenbendazole or add metronidazole to the admission treatment regimen.➢If pyrantel pamoate is used, select a product containing oxantel to control whipworms, except for pregnant or nursing dogs. A single dose of praziquantel (5–10 mg/Kg) can be administered to treat tapeworms.➢Where coccidia are of concern in incoming puppies or dogs, add ponazuril or sulphadimethoxine. If there are concerns about the efficacy of treatment (anthelmintic resistance), examine three fecal samples over 1 to 2 weeks after treatment.➢If funds are available, add selamectin to target fleas, lice, ticks and heartworm prevention, as well as providing some control of hookworms (adults) and ascarids. Puppies and kittens should be administered pyrantel pamoate every 2 weeks, which is usually administered at the time of each vaccination.➢Proper sanitation is essential to reduce the risk of infection to animals and shelter workers.➢Extra care should be taken to remove feces as quickly as possible, especially in the first week following treatment, because of continued shedding of eggs in feces. Feces should be promptly removed, particularly in common areas where dogs are walked or play together, and the area regularly cleaned.➢Hookworm larvae are effectively killed by strong sunlight, 50 ppm aqueous iodine, water above 80 °C, 70% ethanol and Dettol^®^, while ascarid eggs are resistant to most disinfectants. It is extremely important to first remove all feces from pens, runs, and kennels, followed by thorough mechanical cleaning. This will reduce the risk of re-infection and spread to shelter staff [116]. Note standard shelter sanitation with accelerated hydrogen peroxide (spray on surfaces, leave to dry and do not rinse), will eliminate parvovirus, major bacterial and fungal pathogens, but is likely, not effective against hookworm larvae or ascarid eggs. However, general mechanical cleaning is beneficial for clearing all major parasite life stages from the environment (as well as for dermatophyte spores).➢If there are concerns about the spread of *Giardia* spp. or *Cryptosporidium,* dogs should be bathed on the first and last day of treatment because re-infection may occur with cysts and oocysts from the fur [117].➢If anthelmintic resistance is suspected, perform fortnightly fecal egg counts to monitor parasite burdens across the facility. New and recent arrivals should be prioritized. Ultimately the objective should be to screen all animals fortnightly. This could be managed by testing 10% of the population five days a week.➢Treat on exit from the facility either with pyrantel pamoate or ideally, using a 3-day course of fenbendazole (50 mg/Kg of body weight every 24 h). This is essential from a public health perspective. If funds are available, praziquantel (5–10 mg/Kg single dose) should also be administered for tapeworm. Because treatment on exit is aimed at public safety and public relations, treatment should be guided by fecal screening, and where funds are limited, focus these where this is a public health risk with zoonosis.

## 5. Conclusions

It is fairly well established that animals entering shelters carry parasites, and shelters provide not only temporary homes to stray or lost animals, but also an environment that can enhance the spread of GI parasites. Animal shelters should strive to follow the guidelines suggested by the Society of Shelter Veterinarians to ensure the health of animals, working staff, visitors and new animal owners. Integrated approaches (cleaning, disinfection, proper husbandry practices, reducing environmental contamination and periodic fecal examination) should be accompanied with appropriate anthelmintics and regular deworming programs to control and prevent GI parasites in shelter dogs.

## Figures and Tables

**Table 1 animals-08-00108-t001:** Selected reports of prevalence of gastrointestinal parasites in shelter dogs.

Author(s)	Parasite	Prevalence (%)	Methods Used in the Study	Country	Additional Comments
Papini, et al. [31]	*Giardia* spp.	55.3	enzyme-linked immunosorbent assay (ELISA)	Rome, Italy	The relatively high prevalence reported is likely to reflect the sensitivity of the ELISA technique.
Palmer, et al. [32]	Hookworm spp.	10.7	Fecal examination Malachite green staining	Australia	Study included considerable sample size, but use of microscopy for detection of parasites might have led to underestimation of true prevalence.
	*T. vulpis*	3.1			
	*T. canis*	2.4			
	*D. caninum*	0.3			
	*Giardia* spp.	14.4			
	*I. ohioensis*	5.6			
	*I. canis*	1.4			
	*Sarcocystis* spp.	3.2			
	*Cryptosporidium* spp.	0.7			
Mukaratirwa and Singh [33]	Hookworm spp.	53.8	Fecal examination	Durban and Coast, South Africa	Use of microscopy for detection of parasites might have led to underestimation of true prevalence.
	*T. vulpis*	7.9			
	*T. canis*	7.9			
	*G. intestinalis*	5.6			
	*Isospora* spp.	1.3			
Titilincu, et al. [34]	*Cryptosporidium* spp.	37.9	ELISA	Romania	A higher level of *Cryptosporidium* prevalence might be related to use of a sensitive technique (ELISA)
Baharmi, et al. [30]	Hookworm spp.	33.03	Fecal examination, Ziehl-Neelsen trichrome and Iodine staining	Iran	
	*T. vulpis*	8.03			
	*T. canis*	36.6			
	*D. caninum*	10.71			
	*Giardia* spp.	18.75			
	*I. ohioensis*	15.17			
	*Cryptosporidium*	7.14			
	*Taenia* spp.	19.64			
Joffe, et al. [23]	Hookworm spp.	0.81	Fecal examination	Calgary, Canada	
	*T. canis*	12.0			
	*Giardia* spp.	4.2			
Ortuno and Castella [21]	Hookworm spp.	5.3	Fecal examination	Barcelona, Spain	
	*T. vulpis*	11.0			
	*T. canis*	7.5			
	*D. caninum*	0.4			
	*Giardia* spp.	40.6			
	*Isospora* spp.	16.4			
Becker, et al. [35]	Hookworm spp.	4.1	FEC, SNAP^®^ Giardia Test (IDEXX, Westbrook, ME, USA)	Evora, Portugal	Higher prevalence of *Giardia* reflects the higher sensitivity of the SNAP *Giardia* test.
	*T. vulpis*	2.0			
	*T. canis*	0.9			
	*Giardia* spp.	47.0			
	*Isospora* spp.	6.1			
Mahdy, et al. [28]	Hookworm spp.	28.7	Fecal examination	Malaysia	
Ortuño, et al. [36]	Hookworm spp.	3.7	Fecal examination polymerase chain reaction (PCR) for *Giardia* positive samples only	Spain	Shelter protocol was that all dogs received anthelmintics at the time of entry and then every three months thereafter. Use of PCR was likely a contributor to the high prevalence of *Giardia*.
	*T. vulpis*	3.7			
	*T. canis*	7.4			
	*T. leonina*	2.4			
	*Giardia* spp.	63.0			
	*Isospora ohioensis complex*	24.6			
	*I. canis*	6.2			
Alvarado-Esquivel, et al. [25]	*A. caninum*	88.1	Fecal examination and Hematocrit	Veracruz (Mexico)	Dogs with other systemic infections were more likely to have parasitic infestation.
	*T. canis*	45.5			
	*U. stenocephala*	42.6			
	*T. vulpis*	18.8			
	*S. canis*	15.8			
Villeneuve, et al. [10]	*T. canis*	12.7	Fecal Examination and Multiplex PCR	Canada	
	*Isospora* spp.	10.4			
	*Sarcocystis*	4.5			
	*Trichuris vulpis*	4.4			
	*Giardia*	3.5			
	*T. leonina*	3.0			
	*Cryptosporidium*	3.0			
	*U. stenocephala*	2.9			
	*Taenia* spp.	1.6			
Sommer, et al. [26]	*A. canimum*	41.0	Merthiolate-iodine-formalin concentration method, Giardia- Coproantigen	Belgrade, Serbia	134 fecal samples were examined for gastrointestinal parasites and the majority of the dogs were infected with at least one of nine different parasites.
	*G. intestinalis*	45.5			
	*Neospora* spp.	11.2			
	*T. leonina*	9.7			
	*Isospora canis*	8.2			
	*T. vulpis*	6.7			
	*Sarcocystis* spp.	4.5			
	*T. canis*	3.0			

*G. intestinalis* = *Giardia intestinalis*, *A. caninum* = *Ancylostoma caninum*, *U. stenocephala* = *Uncinaria stenocephala*, *T. leonina* = *Toxascaris leonina*, *T. vulpis* = *Trichuris vulpis*, *T. canis* = *Toxocara canis*, *D. caninum* = *Dipylidium caninum*, *I.* = *Isospora*, *S. canis* = *Strongyloides canis*, FEC = Fecal egg count. Note: All datasets were searched from 2005 to 2017 using PubMed and Google Scholar with keywords “prevalence, gastrointestinal parasites, shelter dogs”. Additional relevant articles were identified from references cited in the articles found in the primary search.

**Table 2 animals-08-00108-t002:** Therapeutic agents for animal shelters to control and treat parasitism.

Drug/s Protocol	Target Parasites	Additional Comments
amprolium	*Coccidia*	Significant side effects can occur. Use is off-label in many countries
azithromycin	*Cryptosporidium*	Typically a self-limiting infection and is not treated. However, if treatment is necessary, use every 24 hours until clinical signs resolve
epsiprantel	Tapeworms	Should be avoided in animals younger than 7 weeks and pregnant animals, single dose
emodepside + praziquantel	Roundworms and tapeworms	Emodepside, a relatively newer anthelmintic. Available as a tablet for dogs, although not accessible in all markets
fenbendazole	Ascarids, hookworms, *Trichuris vulpis*, certain tapeworms and *Giardia*	Should be given 3 days consecutively for whipworm and 3–5 days for *Giardia* (efficacy against *Giardia* is controversial)
ivermectin	Ascarids, hookworms and external parasites	Injectable, inexpensive and single dose is sufficient for most of the parasites; should not be used in collie breed and puppies less than 6 weeks old
milbemycin	All helminths	Available in combination with many anti-flea and anticestodal products
moxidectin	Hookworm, heartworm, fleas, mites, and roundworms	Available as spot-on, oral, drench and injectable forms
metronidazole	*Giardia*	Metronidazole eliminates *Giardia* in 2/3rds of dogs and can be combined with fenbendazole for improved efficacy
piperazine	Roundworms	An older heterocyclic compound. Not recommended in combination with pyrantel
ponazuril	*Coccidia*	Registered for use in horses—Use in dogs constitutes off-label use in most jurisdictions
praziquantel	Nearly all tapeworms	Should not be used in younger animals (<4 weeks); a single dose is sufficient
pyrantel	Ascarids and hookworms	Should not be used in combination with piperazine
pyrantel + praziquantel	Ascarids, hookworms, and tapeworms	Addition of praziquantel extends spectrum to tapeworms; should not be used in younger animals (<4 weeks)
pyrantel + praziquantel + febantel	Ascarids, whipworms, hookworms, and tapeworms	More expensive than pyrantel alone; should be avoided in animals younger than 7 weeks and pregnant animals, a single dose for all worms except whipworms
sulphadimethoxine	*Coccidia*	Only approved drug to treat coccidiosis in the USA
selamectin	Hookworms, heartworm, ascarids, fleas, lice, and ticks	Spot-on application. Tolerance and safety margin in dogs with the MDR-1 mutation (e.g., collies) that are sensitive to ivermectin is higher for selamectin than for ivermectin
emodepside + toltrazuril	*Coccidia*	Efficient coccidiocidal in dogs

**Table 3 animals-08-00108-t003:** Antiparasitic drugs and their efficacies in dogs for parasitic treatment.

Drugs	Brand Name	Parasite(s)	Efficacy	References
febantel, pyrantel, praziquantel	(Drontal Plus^®^; Bayer, Ontario, Canada)	Ascarids (*T. cani*; *T. leonina*)	92–100%	[106]
		Ancylostomids	90–100%	
		Taeniidae	73–91%	
fenbendazole	(Panacur^®^; Intervet, Vienna, Austria)	Ascarids	80–100%	
		Ancylostomids	99–100%	
		Taeniidae	90–100%	
mebendazole	(Telmin^®^; Esteve, Cologno Monzese, Italy)	Ascarids	98–100%	
		Ancylostomids	100%	
		Taeniidae	70–90%	
pyrantel, febantel, praziquantel	(Drontal^®^ Plus, Bayer, Ontario, Canada)	*A. ceylanicum*	100%	[107]
Emodepside, praziquantel	Profender^®^ (Kansas, KS, USA)	Ascarids (*T. canis*; *T. leonina*); Whipworms (*T. vulpis*), Ancylostomids (*U. stenocephala*; *A. caninum*),	99.9%	[108]
		Cestodes (*D. caninum*; Taeniidae; *Mesocestoides*)	100%	
emodepside plus toltazuril	(Procox^®^ Tablets for Dogs, Bayer, Leverkusen, Germany)	Ascarids (*T. canis);*	100%	[109]
		Ancylostomids (*A. caninum*; *U. stenocephala*)	99.5–100%	
emodepside plus toltazuril	(Procox^®^ Tablets for Dogs, Bayer, Leverkusen, Germany)	*Isospora* spp.	90.2–100%	[110]

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
