# Peer review of "Gastrointestinal Parasites in Shelter Dogs: Occurrence, Pathology, Treatment and Risk to Shelter Workers"

_animals, 2018, doi:10.3390/ani8070108_

Round 1

Reviewer 1 Report

Review of Raza et al (resubmitted version) for Animals Jun 2018

In rewriting the manuscript the authors have addressed all points raised in the initial review and with minor further changes I would be very happy for the paper to be accepted.

Specific comments:

Line 35 “ingestion” of infective stages is not the main means by which infections with A. caninum are acquired.

Line 101 delete “route”

Table 1:

Please check formatting again.  There is an extra line below study ref 33 (and ref 10), and a space below the first parasite named (hookworm).

For study ref 36, the text still refers to Cystoisospora.

Line 155 soil is important for providing protection to hookworm larvae, rather than their eggs.  The eggs are mainly developing in faecal material.

Line 368 The dispersal stage for Cryptosporidium is the oocyst rather than a cyst.

Table 2:

Still no mention of milbemycin.  This anthelmintic is now in combination with many flea products.

Table 4:

First bullet point – praziquantel is misspelt.

Author Response

We are really thankful to the reviewer for potential inputs to improve the manuscript for publication.

1.      Line 35 “ingestion” of infective stages is not the main means by which infections with A. caninum are acquired. 

Response: Thanks for highlighting this, we have modified the text to “ingestion or skin penetration”

2.      Line 101 delete “route”

Response: Deleted as suggested

Table 1: Please check formatting again.  There is an extra line below study ref 33 (and ref 10), and a space below the first parasite named (hookworm).

For study ref 36, the text still refers to Cystoisospora.

Response: Extra line below study ref 33 and the space below the first parasite named (hookworm) has now been removed. Also, Cystoisospora has been changed to Isospora.

3.      Line 155 soil is important for providing protection to hookworm larvae, rather than their eggs.  The eggs are mainly developing in faecal material.

Response: The text has been modified by replacing “eggs” with “larvae”.

4.      Line 368 The dispersal stage for Cryptosporidium is the oocyst rather than a cyst

Response: The reviewer is correct here. We have modified the cyst to oocyst.

5.      Table 2: Still no mention of milbemycin.  This anthelmintic is now in combination with many flea products.

Response: Milbemycin has now been added to table 2.

6.      Table 4: First bullet point – praziquantel is misspelt.

Response:  “praziquantal” has been correctly spelled now to “praziquantel”

Reviewer 2 Report

The quality of the manuscript in this new version has notably improved, but there are still some points that need to be addressed/corrected. In addition, there are still some references that are wrongly cyted in the text (I highly recommend to the authors a careful revision of all the references (not only those marked in my comments) in the text before submitting the final version).

Please see the specific comments in the attached file.

Author Response

We are really thankful to the reviewer for potential inputs to improve the manuscript for publication.
